# PLGA-Based Nanoparticles for Neuroprotective Drug Delivery in Neurodegenerative Diseases

**DOI:** 10.3390/pharmaceutics13071042

**Published:** 2021-07-08

**Authors:** Anthony Cunha, Alexandra Gaubert, Laurent Latxague, Benjamin Dehay

**Affiliations:** 1Université de Bordeaux, INSERM U1212, CNRS UMR 5320, ARNA, ARN: Régulations Naturelle et Artificielle, ChemBioPharm, 146 rue Léo Saignat, F-33076 Bordeaux, France; anthony.cunha@u-bordeaux.fr; 2Univ. Bordeaux, CNRS, IMN, UMR 5293, F-33000 Bordeaux, France

**Keywords:** PLGA nanoparticles, neurodegenerative diseases, drug delivery, central nervous system, neuroprotective drugs

## Abstract

Treatment of neurodegenerative diseases has become one of the most challenging topics of the last decades due to their prevalence and increasing societal cost. The crucial point of the non-invasive therapeutic strategy for neurological disorder treatment relies on the drugs’ passage through the blood-brain barrier (BBB). Indeed, this biological barrier is involved in cerebral vascular homeostasis by its tight junctions, for example. One way to overcome this limit and deliver neuroprotective substances in the brain relies on nanotechnology-based approaches. Poly(lactic-*co*-glycolic acid) nanoparticles (PLGA NPs) are biocompatible, non-toxic, and provide many benefits, including improved drug solubility, protection against enzymatic digestion, increased targeting efficiency, and enhanced cellular internalization. This review will present an overview of the latest findings and advances in the PLGA NP-based approach for neuroprotective drug delivery in the case of neurodegenerative disease treatment (i.e., Alzheimer’s, Parkinson’s, Huntington’s diseases, Amyotrophic Lateral, and Multiple Sclerosis).

## 1. Introduction

Neurodegenerative diseases (NDD) represent a major societal issue. For example, in 2015, more than 46.8 million people worldwide were affected by dementia [1] and over 6 million by Parkinson’s disease (PD) [2]. This number is increasing each year due to, in particular, the population aging. NDDs are a complex group of diseases with no curative treatments for the five most common and known of them: Alzheimer’s disease (AD), Parkinson’s disease, Huntington’s disease (HD), Amyotrophic Lateral Sclerosis (ALS), and Multiple Sclerosis (MS) (Table 1). These incurable and incapacitating diseases lead to the progressive degeneration and death of nerve cells, causing troubles related to motor disabilities or mental dysfunction (called dementias), considerably worsening patient quality of life.

Therefore, the development of chronic treatments for NDDs both effective and easy to administer remains a major challenge due to the necessary blood-brain barrier (BBB) crossing to get drugs into the brain. This barrier filters and controls the passage of foreign blood substances, potentially dangerous molecules or pathogens, and prevents them from passing freely from the blood into the extracellular fluid of the brain’s gray matter. In addition, the BBB protects the brain cells by regulating their environment (cerebral vascular homeostasis, pH) against hormone and neurotransmitter variations, for example. Thus, it offers strong resistance to the ion movement, allowing to maintain the brain chemical balance and adjusting this environment to guarantee a perfect transmission of signals between neurons. In addition, the existence of tight junctions (TJs) between each endothelial cell greatly limits the cerebral passage of numerous compounds (98% of the substances cannot cross it) [3,4].


pharmaceutics-13-01042-t001_Table 1Table 1Non-exhaustive description of most known neurodegenerative diseases.DiseasePrevalence (Per 100,000 Person)SymptomsAffected AreasAD511–690 [5]Memory impairment, changes in thinking, judgment, language, behavioral changes, etc.Cortex, hippocampus, brainstemPD100–200 [6]Rest tremors, slowness of movement, decrease of spontaneous mobility, muscular stiffness, etc.Basal ganglia, cortexHD5.96–13.7 [7]Motor disorders, breathing difficulties, speech and swallowing disorders, etc.Striatum and other basal ganglia regionsALS3.92–4.96 [8]Progressive muscle paralysis, muscle atrophy, spasticity, breathing and swallowing disorders, etc.Motor cortex, spinal cordMS35.87–35.95 [9]Numbness in a limb, vision problems, electric shock sensations in a limb or the back, movement problems, etc.Brain, spinal cord, optic nerve


One way to overcome these drawbacks relies on nanotechnologies. This approach presents various advantages such as drug protection, targeting, controlled release, or interest in NDD treatment, the BBB crossing. A wide range of pharmaceutical nanocarriers was developed, including liposomes, solid lipid nanoparticles, micelles, dendrimers, or nanoparticles (NPs) [10,11].

Thanks to their physicochemical characteristics, such as their size, biocompatibility, or low cytotoxicity, polymeric NPs represent very interesting tools for delivering neuroprotective agents for NDD treatments [12]. Indeed, their ability to cross biological barriers, their versatility (encapsulation of either hydrophilic or hydrophobic drugs), or their targeting ability (drug delivery to a specific body site decreasing the off-target toxicity) explains the increasing interest for this approach. Among the polymers used for NP formulation (natural or synthetic), the most widely used is the poly(lactic-*co*-glycolic) acid (PLGA). This polymer has a double interest in the NDD treatment for its acidifying [13], particularly for PD, and drug encapsulation properties [14].

In the following review, we propose a non-exhaustive summary of the last decade’s advances in delivering neuroprotective substances via PLGA NPs for NDD treatment. A focus will be made on the existing neuroprotective drugs such as phytol, rhynchophylline, or curcumin to go to the potential new neuroprotective candidates, benefiting from the PLGA NP vectorization.

## 2. PLGA NPs

Nowadays, PLGA NPs are a booming topic, especially for the development of NDD treatments (more than 50 articles during the last 5 years on Scopus using “PLGA”, “nanoparticles”, “neurodegenerative disease” as keywords). PLGA, a commercially available synthetic copolymer obtained from lactic and glycolic acid, is approved by the US regulatory agency (Food and Drug Administration: FDA) and the European Medicine Agency (EMA). To date, not based on PLGA but poly-lactic acid (PLA), paclitaxel-loaded PEG-PLA micelles have reached the market in South Korea, India, and Indonesia (Genexol^®^ PM). It is currently undergoing Phase III clinical trials for access to the EU and US markets. For the NPs, only one Phase II clinical trial based on PEG-PLGA/PLA-PEG NPs (BIND-014) for metastatic castration-resistant prostate cancer was reported [15]. However, in the case of NDDs treatment, no PLGA NPs are currently on the market or in clinical trials, but are currently only at the preclinical stage [16]. Indeed, several pre-clinical studies based on drug loaded nano-objects are in progress, including delivery of curcumin, levodopa, cholesterol or rapamycin for AD, PD, HD and MS treatment. The advantage of the PLGA NPs approach relies on their potential for drug encapsulation, excellent biocompatibility, and biodegradability [17]. PLGA degradation by hydrolysis of its ester bonds in aqueous media releases its two constitutive monomers, which are naturally produced under physiological conditions by several metabolic pathways [18,19]. Thanks to this property, PLGA was reported as an active substance, notably for the treatment of PD since lactic acid and glycolic acid decrease lysosomal pH [13,20]. For example, studies on pathological PD models demonstrated that PLGA NPs reacidified defective lysosomes to basal level (pH 4), restored lysosomal deleterious effects, due to PD-linked mutations, and overcame lysosomal acidification impairments. The acidification also supports mitochondrial membrane potential and neuronal survival in several models related to mitochondrial dysfunctions.

### 2.1. PLGA NP Formulation and Optimization

These last years, a wide variety of preparation techniques for PLGA NPs were reported in the literature, i.e., single and double emulsion solvent evaporation, nanoprecipitation, coacervation and spray-drying [14,21]. The single and double emulsion solvent evaporation methods are the most frequently used: PLGA is dissolved into an organic phase emulsified with an aqueous medium containing surfactants or stabilizers prior to solvent evaporation and NP formation, corresponding to a single emulsion solvent evaporation method [22]. The single method involves a two-phase process (oil in water O/W or water in oil W/O) while three-phase are used with the double emulsion solvent method (O/W/O or W/O/W). The main drawback of these two techniques is the use of shear stress during the homogenization step leading to low protein encapsulation efficiency. To overcome this drawback, the nanoprecipitation method may be beneficial. In that case, the organic solvent containing PLGA is added dropwise to the aqueous medium, leading to NPs formation by a rapid diffusion of the miscible solvent [23]. Nevertheless, the previous methods are not the most appropriate for industrial scale-up, unlike spray drying, which transforms liquid substances (sprayed in a thin stream of heated air) into powders [14].

Depending on the process used, amphiphilic, hydrophobic, or hydrophilic drugs can be either loaded inside the NP core, trapped among the polymer chains, or adsorbed on the NP surface (leading to burst release and potential off target effects). Even though a low encapsulation efficiency characterizes PLGA NPs, in particular for hydrophilic drugs, there is room for improvement by playing with the molecular weight (i.e., the higher the molecular weight, the larger the NPs) or the poly(d,l-lactic acid) (PLA)/poly (d,l-glycolyc acid) (PGA) ratio [24]. The latter parameter is directly related to the crystallinity degree and the melting point of PLGA copolymers, and drives the capacity of polymers to undergo hydrolysis. Indeed, PLA exhibiting methyl side groups is more hydrophobic than PGA, leading to a slower degradation of PLGA NPs possessing a high PLA ratio [17].

Depending on the biomedical application, other parameters must be considered for NP optimization such as particle size, polydispersity index (PdI), zeta potential (ζ potential), drug loading, or encapsulation efficiency. It was reported that a diameter lower than 200 nm enables an increase of the in vivo half-life and a better membrane passage, which are crucial for biomedical applications [25,26]. If the NP diameter is essential, the size distribution, characterized by the Polydispersity Index (PdI), is just as important: the smaller the PdI, the more homogeneous the population of NPs. Values of 0.2 and below are commonly deemed acceptable in practice for biomedical applications of polymer-based NPs [27]. To ensure a moderate colloidal stability of these nano-objects, a ζ potential below −30 mV or above +30 mV is required to avoid coagulation or flocculation phenomena [28]. Moreover, the NP charge also impacts their biological effects: positively charged NPs increase cellular uptake and cytotoxicity in nonphagocytic cells while negatively charged NPs enhance cytotoxicity in phagocytic cells [29].

Modulation of NP functionalization enables designing NP platforms to address many issues, especially in the diagnostic, the active targeting, or the shielding fields (Figure 1) [30,31]. Regarding the diagnostic approach, probes (containing fluorescent moieties, isotopes, etc.) are introduced inside the NP or at the surface, allowing the detection by different imaging techniques: near-infrared, fluorescence, positron-emission tomography, or single-photon emission computed tomography [30]. Another aspect is proper targeting to ensure the drug delivery directly on site. Ligands such as proteins, polysaccharides, peptides, aptamers, or small molecules are promising candidates in this approach [32]. For instance, tween 80, also known as polysorbate 80, can adsorb apolipoprotein E onto PLGA NPs enabling the binding of lipoprotein receptor-related proteins (LRPs), thus facilitating the BBB crossing [33]. This hydrophilic nonionic surfactant and emulsifier is usually used at 1% (*w*/*v*) to stabilize aqueous formulations for parenteral administration. It is speculated that weak interactions control the formation of an adsorbed monolayer of polysorbate 80 onto the nanoparticle surface [34]. As a coating, polysorbate 80 presents some clinical advantages such as decreasing the drug dosage (therefore reducing the potential side effects) or increasing its viability. Nevertheless, liver and renal toxicity, hypersensitivity or erythema need to be monitored to ensure treatment safety [35] About intranasal administration, the critical point relies on the residence time of the drug. This problem can be addressed by using mucoadhesive polymers, such as chitosan or specific ligands such as lectins [36]. A better binding to the nasoepithelial surface increases the residence time and thus the drug bioavailability. The last point to consider is the shielding of the nano-objects to reduce aggregation, opsonization, phagocytosis, and immune clearance, which may otherwise decrease their circulation time and their targeting efficiency. For this purpose, NPs are coated with several molecules such as carbohydrates, proteins, or lipids, knowing that PEGylation remains the main strategy to improve drug efficiency and gene delivery [37,38,39].

### 2.2. Administration Routes

Due to their incredible modularity and small size, PLGA NPs present numerous advantages for NDD treatments via different available administration routes [40]. Stereotaxic surgery overpasses the BBB and thus directly delivers the drugs without peripheral drug inactivation [41]. The main advantage of this approach is to be as precise as possible on the target despite the technic invasiveness. Otherwise, other techniques such as enteral, parental, and intranasal routes represent good alternatives. The first two require crossing the BBB through various mechanisms (simple diffusion, receptor, adsorption mediated endocytosis, or carrier-mediated transport), depending on NP functionalization, with a first pass effect decreasing the drug concentration on site in the case of enteral route [42,43]. The last one (intraneuronal absorption by a direct nose to brain delivery of the NP platforms), allows a rapid and important active substance passage into the systemic circulation without first-pass metabolism avoiding drug degradation. It offers the advantage of a better patient compliance, due to the low invasiveness of the daily-based administrations, and therefore represents a promising option for NDD treatment [44].

Two types of strategies can be considered with PLGA NPs for NDD therapy: as active substances by themselves and as cargos for neuroprotective drugs.

## 3. PLGA NPs for Neuroprotective Drug Delivery in Neurological Disorder Therapy

Many neuroprotective medicines cannot reach the brain due to the lack of drug-specific transport systems through the BBB. The development of new strategies based on PLGA NPs to enhance brain drug delivery is of great interest in NDD therapy [45]. Indeed, even if different pathologies are included in NDD, they all share common mechanisms such as mitochondrial or oxidative injury, neuro-inflammation, apoptosis, and protein aggregation, which contribute to neuronal loss mainly by the intrinsic mitochondrial apoptotic pathway (Figure 2) [46,47]. The mitochondria complexes and mitochondrially located monoamine oxidase B (MAO-B) are involved in the nitrogen and reactive oxygen species (ROS) production. The dysfunction of these pathways is one characteristic of NDD as the neuro-inflammation for AD, PD, and MS. Neuro-inflammation is a defense mechanism that initially protects the brain by removing or inhibiting diverse pathogens. In the case of NDD, it involves abundant activated microglia inhibiting amyloid-beta (Aβ) phagocytosis contributing to plaque accumulation. These mechanisms are targets of choice for treating protein aggregation diseases (proteinopathies) such as AD, PD, HD, ALS, and MS.

To act on the mechanisms involved in NDDs, the new trend relies on the encapsulation of specific active compounds (natural or synthetic) into PLGA NPs (Table 2).

### 3.1. Alzheimer’s Disease

AD is the most common neurodegenerative disease responsible for a progressive decline in cognitive functions. The association of two neuropathological brain lesions characterizes this pathology: extracellular Aβ protein and intracellular hyperphosphorylated tau protein deposits [48,49]. Over time, neurodegeneration occurs in the brain regions related to memory and language, from the hippocampal region to the entire cerebral cortex, explaining the progression of the symptoms to aphasia, apraxia, visio-spatial, and executive function disorders [50]. The causes of protein aggregation, into amyloid deposits and neurofibrillary degeneration, are still unknown, but genetic and environmental factors may contribute to their appearance [51]. The protein aggregation may, in turn, induce oxidative stress (OS) leading to neuroinflammation [52]. OS results from an imbalance between the production of free radicals and their detoxification rates by the SuperOxide Dismutase enzyme (SOD) especially.

There is currently no cure for AD, only palliative strategies (tacrine, rivastigmine, etc.) to limit its progression and improve the patient’s quality of life, opening the doors for the nanotechnology approach and more especially the PLGA NP one (Table 2).

#### 3.1.1. Thymoquinone

As previously said, AD is linked to protein (i.e., Aβ) aggregation which may, in turn, induce oxidative stress leading to neuro-inflammation. Yusuf et al. designed PLGA NPs loaded with thymoquinone (TQ), a phytochemical compound, to act on this aspect in animal models due to its potent antioxidant and anti-inflammatory properties (Figure 3) [53,54]. Male albino mice administered with Streptozotocin (SZT) (mimicking AD oxidative stress by a decrease of SOD activity) were treated with TQ loaded PLGA NPs, coated with polysorbate 80 (P-80-TQN). These NP were prepared by the single-emulsion solvent-evaporation method, leading to spherical nano-objects with an average size of 226 ± 4.6 nm and −45.6 ± 2.6 mV as ζ-potential. In vitro, TQ release assessment was characterized by an initial burst for 2 h (drug diffusion from the NP surface and by the beginning of PLGA matrix erosion) followed by a longer sustained release (stabilized dipole-dipole interactions taking place between TQ and PLGA components). The P-80-TQ NPs successfully crossed the BBB by endocytosis through LDL receptors (mediated by the polysorbate coating). Once on-site, these systems significantly affected SOD activity (increase) from day 7 to day 28. Concurrently, an animal and cognition test (Despair test) was undertaken to highlight a meaningful improvement after NP injection, further confirming their beneficial role. Finally, histopathological examinations of brain hippocampal tissues, before and after P-80-TQ NP treatment (5 mg.kg^−1^ equivalent), showed a drastic size reduction of the protein aggregate generated by the STZ-induced AD in mice.

#### 3.1.2. Huperzine A

Meng et al., developed an intranasal drug delivery system (DDS) as PLGA NPs loaded with Huperzine A (HupA) (Figure 3) co-modified with lactoferrin (Lf) to enhance nose-to-brain delivery and N-trimethylchitosan (TMC) for easier mucoadhesion [55]. HupA possesses neuroprotective effects especially in the treatment of AD. Still, it displays a poor brain selectivity resulting in several side effects (including nausea, diarrhea, vomiting, …), justifying the need for a more specific DDS [56]. NPs were formulated by the single-emulsion solvent-evaporation method with maleimide-TMC, allowing the further conjugation of thiolated Lf on the maleimide moiety leading to the Lf-TMC-NPs. No cytotoxicity was observed for free HupA and HupA-loaded NPs concentrations ranging from 12.5 to 25 µg.mL^−1^ against human bronchial epithelial cell lines (16HBE) used as a model of nasal mucosa cells. The mucoadhesive assessment was evaluated by mucin adsorption onto NPs with a binding efficiency up to 86% compared to unmodified PLGA NPs (32%). Lf-TMC NPs achieved a boost release of HupA in vitro within 4 h followed by a sustained release of 74.5% at 48 h. The cellular uptake of NPs was assessed using 16HBE and SH-SY5Y (human neuroblastoma cell line) cells as brain cell models, using nile red-loaded NPs. They were taken up in both cell lines with a significantly higher fluorescence intensity for Lf-TMC NPs than unmodified NPs or even TMC-NPs. The same trend was explicitly observed in vivo after intranasal administration of DiR-loaded NPs in mice. A strong signal was obtained with TMC-NPs in the brain, attributed to a better nose-to-brain drug delivery and mucoadhesion provided by TMC. Furthermore, an even larger Lf-TMC NPs was localized in the brain, suggesting a Lf-mediated transport mechanism. Finally, following the intranasal administration of HupA loaded NPs, pharmacokinetics exhibited a minimal elimination rate associated with a greater bioavailability of Lf-TMC NPs, particularly in the olfactory bulb and the memory-related hippocampus.

#### 3.1.3. Rhynchophylline

Rhynchophylline (RIN) is a spirocyclic alkaloid isolated from Uncaria species that exhibits numerous pharmacological activities, including neuroprotective effect (Figure 3) [57]. In the case of AD, RIN inhibits soluble Aβ-induced hyperexcitability of hippocampal neurons. In 2020, the first study on the design and development of brain-targeting therapy for AD using RIN administration was published by Xu et al. [58]. Tween 80 coated methoxy poly (ethylene glycol) PLGA NPs were prepared by nanoprecipitation technique to enhance the pharmacological activity of RIN and targeting purposes. No hemolysis occurred using T80 coated RIN loaded PLGA NPs, suggesting that these NP suspensions were safe to use. Internalization into bEnd.3 cells was confirmed by confocal laser scanning microscopy, using DiD fluorophore-loaded PLGA NPs; T80 coated NPs exhibited higher permeation. To go further, an in-vitro BBB model, with bEnd.3 cells, was developed to study the crossing of such nano-systems and demonstrated their higher transport compared to free RIN or uncoated RIN loaded NPs. The benefits of using T80 for targeting brain was then highlighted in vivo in healthy C57BL/6 mice. Finally, survival rate of PC12 cells injured by Aβ_25–35_ was improved by incubation with T80 RIN NPs, and cell apoptosis was significantly reduced. These results confirmed that encapsulation did not affect RIN neuroprotective effects, demonstrating the great potential of loaded PLGA NPs for NDD therapy.

### 3.2. Parkinson’s Disease

PD is the second most common NDD after AD. Three characteristic motor symptoms are associated with Parkinson’s syndrome: tremor, akinesia (slowness of movement), and limb rigidity [59]. The disease hallmarks are defined by a selective neuronal dopaminergic loss in specific brain regions, and deposits of misfolded proteins through the presence of α-synuclein aggregates (Lewy bodies, LB). Other pathophysiological symptoms include altered dopamine metabolism, impaired mitochondrial function, oxidative stress, and inflammation. Nowadays, no cure is available for AD, but symptomatic treatments, such as L-dopa (dopamine precursor), can reduce the motor symptoms. This lack of therapeutic alternatives paves the way for the nanotechnology approach, especially using the PLGA system, since this compound was proved biologically active in PD treatment [13,20]. Combining PLGA with active drugs could thus constitute a good option to achieve a synergistic action on PD (Table 2).

#### 3.2.1. Schisantherin A

Schisantherin A (SA) is a dibenzocyclooctadiene with neuroprotective activity against MPP(+) (or 1-methyl-4-phenyl pyridinium, a toxic molecule interfering with oxidative phosphorylation in mitochondria), thus a potential candidate for neuron loss treatment in PD (Figure 4) [60]. In 2017, Chen et al., prepared spherical SA-NPs (ø: 70.6 ± 2.2 nm), by flash nanoprecipitation method, stable for one week at room temperature [61]. For oral administration purposes, the in vitro drug release studies were conducted in a simulated gastric fluid with pepsin (SGF, pH 1.2), a simulated intestinal fluid with trypsin (SIF, pH 6.8), and PBS at physiological pH (7.4). All conditions gave an almost similar release pattern with a burst release (25%) in the initial 1 h followed by a sustained release reaching 90–94% at 48 h. DiO and DiI fluorophores were loaded into the NP to form FRET-NPs of similar sizes concerning SA-NPs and were stable for more than four weeks, which is required for cellular uptake studies. MDCK cells were incubated with the fluorescent NPs for 2 h and only a moderately decrease of the FRET signal was observed, indicating a slow intracellular release. The epithelial transport of NPs was also assessed with the same FRET-NPs and showed that a fraction of intact NPs was transported from the apical to the basolateral side. In vivo biodistribution studies in adult zebrafish after oral administration for 2 h showed a fast absorption of NPs into the blood circulation before reaching the brain. Similarly, after oral administration of SA-NPs to rats, they directly reached the systemic circulation with a decreased SA elimination rate in plasma (vs. control) and even more in the brain, suggesting specific brain targeting properties of this system. Finally, the neuroprotective effect of SA-NPs was evaluated using an MPTP-induced zebrafish model of PD; a decrease of the MPTP-induced neurotoxicity (vs. control), involving the Akt/Gsk3β signaling pathway activation, was observed.

#### 3.2.2. Puerarin

Puerarin (PU) is a flavonoid glycoside, widely used in China, extracted from the root of the leguminous plant, Pueraria lobata, and Thomson Kudzuvine Root (Figure 4). Puerarin displays a series of beneficial activities notably neuroprotection, antioxidant, etc. [62]. In 2019, Chen et al., described the development of PU-loaded PLGA NPs to treat PD while enhancing half-life (pharmacokinetic study in rats) and bioavailability of the drug [63]. Homogeneous suspensions (ø: 88.4 ± 1.7 nm and PdI: 0.047 ± 0.007) of NPs were prepared by nanoprecipitation. Cytotoxicity and internalization of this formulation were investigated in MDCK cells after 24 h and 48 h. Overall, PU-loaded PLGA NPs exhibited no significant toxicity, and the uptake was confirmed. For oral administration purposes, the in vivo drug release studies were conducted in adult zebrafish; an effective brain uptake was observed using FRET-NPs (DiO and DiI fluorophores). In a dose-dependent manner (from 10 to 50 µM), PU NPs prevented cell death (SH-SY5Y cells) by decreasing MPP+ neurotoxicity and mitochondrial oxidative stress. Finally, PU-loaded NPs improved behavioral deficits, TH-positive neuron loss and associated mobility impairments (i.e., bradykinesia, fall latency, number of falls, average travel distance or speed), and restored dopamine level and MPTP-mediated neurotoxicity in mice.

#### 3.2.3. Rasagiline

Rasagiline (RA) is a selective irreversible second-generation MAO-B inhibitor, which can be used both as monotherapy and as co-adjuvant therapy in combination with levodopa to treat PD (Figure 4) [64,65]. RA efficacy was evaluated using the Unified PD Rating Scale (UPDR). It showed that patients taking the lowest dose at the earliest possible time had slightly better benefic effects at 18 months than patients who received this dose only at 9 months [66]. To overcome poor solubility issues, low bioavailability and half-life, Bali and Salve designed transdermal films of gellan gum on which RA mesylate-loaded PLGA NPs were embedded [67]. This administration route was a novel approach for the sustained release of NPs. The nano-objects were prepared by double emulsion solvent evaporation technique and characterized by DLS (ø: 221.7 ± 5.7 nm) and field emission scanning electron microscopy (FE-SEM). A relatively high PdI (0.388 ± 0.86) was observed, suggesting a heterogeneous suspension. An in vivo study in Wistar rats highlighted an enhanced brain bioavailability using RA-PLGA NPs. Transdermal films slowed down the initial drug release rate followed by a constant release for more than 72 h compared to intravenous or oral administration. The sustained release of RA restored presynaptic depletion of dopamine, inhibited MAO-B enzyme, and prevented neuronal damage caused by oxidative stress in PD model. Finally, gamma scintigraphy study was performed to evaluate BBB permeation and targeting efficiency using ^99m^TC-loaded PLGA NPs embedded transdermal film. It was observed that PLGA NPs are targeted from dorsal part toward the brain region of rats. Overall, this study appeared promising for the development of an easy-to-administer DDS able to cross the BBB and target the brain.

### 3.3. Huntington’s Disease

Described in 1872 by George Huntington, HD is a genetic and hereditary disease, due to the mutation of the gene IT15, resulting in the repeat expansion of CAG trinucleotide, coding for the huntingtin protein (mHtt) [68]. Symptoms of HD often appear between the ages of 30 and 50 years old with the onset of progressive motor (involuntary jerking/writhing movements known as chorea, rigidity, muscle contracture, etc.), behavioral (dementia, difficulty of organization and learning) and psychiatric disorders (feelings of irritability, insomnia) [69]. These troubles are associated with the death of striatal neurons in the cortex, the striatum hippocampus and other brain regions such as the thalamus, subthalamic nucleus, and substantia nigra pars reticulate in advanced cases of HD [70,71]. The mutation of the gene IT15 leads to an accumulation of mHtt proteins and formation of toxic neuronal intranuclear inclusions (NIIs). The only drug approved by the regulatory agencies is the tetrabenazine (TBZ) for the symptomatic relief [72]. Research in the area is thus very interesting to expand the therapeutic arsenal, especially via the PLGA NP approach (Table 2).

#### 3.3.1. Oligonucleotides QBP1, NT17, and PGQ_9_P^2^

Published in 2019, Joshi et al., designed peptide-loaded PLGA NPs coated with polysorbate 80 to act on mHtt aggregation. QBP1, NT17, and PGQ_9_P^2^ are oligonucleotides exhibiting great inhibitory potential against mutant huntingtin protein aggregation [73,74]. In this study, nanoprecipitation peptide-loaded PLGA NPs were prepared to obtain NPs from 158 to 180 nm in diameter with a low PdI (<0.100) and a ζ potential between −23.3 and −27.5 mV. Polysorbate 80 coating enabled an active targeting to the brain, while improving peptide bioavailability and half-life. Overall, in vitro studies performed using Madin-Darby canine kidney (MDCK) cells suggested that transcytosis was facilitated by absorption of apolipoprotein E (ApoE) and low-density lipoprotein (LDL) receptors. Neutral red PLGA NPs were used to conduct in vivo study in healthy mice and confirmed the higher concentration in the brain of polysorbate 80 coated NPs compared to uncoated NPs. The peptide-loaded PLGA NPs displayed dose dependent inhibition of mHtt aggregation in both Neuro 2A and PC12 cell models. Finally, using Drosophila model of HD, their ability to improve motor performance was evaluated; after feeding larvae and adult flies with high NT_17_ and QBP1 concentrations; higher crawling and climbing activities were observed, compared to the control group.

#### 3.3.2. Cholesterol

Cholesterol is a lipid of the sterol family produced by cells and found in food. It is an essential element for synthesizing many hormones, neurotransmitter release, membrane structure, vesicle assembly and fusion. Recently, cholesterol metabolism in the central nervous system of HD patients was described as significantly altered, leading to brain malformations and impaired cognitive functions (Figure 5) [75,76]. A study by Valenza et al., reported the delivery of cholesterol using PLGA NPs to treat HD [77]. Homogenous and monomodal suspensions (ø: 200–300 nm, PdI: 0.09–0.3, dose-dependent) of cholesterol-loaded NPs were prepared with various cholesterol concentrations (1, 5, to 10 µM). In vitro studies showed an initial burst release during the first days followed by a constant release (5–10 days), and an internalization into neuron and astrocyte models of HD. In vivo assays in R6/2 mice highlighted that cholesterol PLGA NPs crossed the BBB and restored electrophysical phenotypes such as synaptic transmission in striatal medium-sized spiny neuron impairment. Finally, cholesterol supplementation did not significantly improve behavioral issues (locomotion, resting time, rearing) but slowed down the disease progression. Using a novel object recognition test for cognitive task evaluation, the cholesterol treated mice group appeared to benefit from the drug with resolved memory deficits.

Even if the therapeutic arsenal for HD treatment is limited to TBZ, the current studies open the way to new promising tools.

### 3.4. Amyotrophic Lateral Sclerosis

ALS, also known as Lou Gehrig’s disease, is a NDD involving the motor neurons (motoneurons) located in the anterior horn of the spinal cord and the motor nuclei of the last cranial nerves. ALS results in a progressive paralysis of the muscles responsible for voluntary motor skills. The symptoms go from muscle twitching, cramping, stiffness, weakness, involuntary jerking movements, tremors, to phonation (the production of sounds) and swallowing [78]. No curative treatment is available, and only riluzole is used to delay the disease evolution [79]. New developments of therapeutics for ALS treatment would be very beneficial (Table 2).

#### PHA-767491

PHA-767491 (PHA) is the first cell division cycle seven kinase inhibitor described as a molecule preventing neurodegeneration, especially in the treatment of ALS (Figure 6) [80]. PHA prevents phosphorylation by CDC7 of TDP-43, a nuclear protein encoded by TARDBP gene-regulating several RNA processes as transcription, mRNA transport, and microRNA biosynthesis. To address low permeability, rapid metabolism, and unspecific distribution issues, Rojas-Prats et al., used PHA-loaded PLGA NPs [81]. They were prepared by nanoprecipitation method and observed by SEM microscopy using gold-coated NPs. The obtained nano-objects exhibited a 141–155 nm diameter with a low PdI (≤0.15) characteristic of a monodisperse suspension. Encapsulation of PHA was controlled and confirmed by imaging PHA fluorescence using confocal laser scanning microscopy. An encapsulation efficiency of 12–18% and a drug loading of 2–4% were determined by HPLC. Using porcine lipid to emulate the human BBB, PHA crossing was enhanced, being loaded in the PLGA NPs compared to free PHA. Finally, in vitro assays, in the SH-SY5Y cell model of TDP-43 phosphorylation, highlighted a significant protection of neuronal cells from death and decreased TDP-43 phosphorylation.

### 3.5. Multiple Sclerosis

Multiple sclerosis (MS) is a chronic autoimmune inflammatory disease and NDD that attacks the CNS, particularly the brain, nerves, and spinal cord [82]. It involves various pathogenic mechanisms leading to inflammatory infiltrates (T cells, B cells, macrophages) and alteration of the myelin, which forms a protective sheath around the nerve endings, and thus of nerve impulse transmission. Symptoms vary depending on the localization of the attacks: limb numbness, vision troubles, electric shock sensations in a limb or the back, movement dysfunctions, etc. Nowadays, over a dozen therapeutic agents reducing the number of attacks and delaying MS progression are approved by the regulatory agencies. A few of them may be vectorized by PLGA NPs to treat some MS symptoms (Table 2) [83].

#### 3.5.1. Interferon-Beta-1a

Interferon-beta-1a (IFN β-1a) is a naturally occurring protein produced by various cell types including fibroblasts and macrophages. Binding specific receptors exert its biological effect on the human cell surface to slow down MS progression [84]. To decrease drug dosage and improve its controlled release, Fodor-Kardos et al., designed IFN β-1a-loaded PEG-PLGA NPs by double emulsion solvent evaporation method with great encapsulation efficiency (95.9%) [85]. According to the used technic: 165 nm (PdI: 0.093) by DLS or 50–120 nm by SEM, differences in the NP size were observed by SEM, linked to the hydration of the sample. BSA loaded NPs were also prepared as a model for release kinetics comparison: a significant slower release was observed in the case of PEG-PLGA NPs compared to simple PLGA NPs. Cyanine 5 amine fluorescent dye was conjugated to the NPs to confirm the cellular uptake and the absence of cytotoxicity in vitro in hepatocytes. Finally, in vivo toxicity studies were performed on healthy Wistar male rats. In the first step, the loaded NPs exerted some toxicity due to the fabrication process and potential remains of solvent, while the unloaded ones were safe to use. In the second step, the formulation process was modified to nanoprecipitation, however mild toxic signals were observed for blank PLGA and PEG-PLGA NPs. To go further, a new formulation process will be required to obtain IFN β-1a-loaded PEG-PLGA NPs without or with low in vivo toxicity allowing to investigate the neuroprotective effect of the system.

#### 3.5.2. Leukemia Inhibitory Factor

Leukemia Inhibitory Factor (LIF) is a neural stem cell growth factor belonging to the interleukin-6 family of cytokines. LIF plays an important role in different biological processes, including differentiation of leukemia cells, inflammatory response, neuronal development, etc. [86]. Recent evidence has demonstrated a crucial role of LIF in neuroprotection, axonal regeneration as well as the prevention of demyelination [87]. Preliminary studies to design and prepared PLGA-NG2 NPs, by double emulsion solvent preparation method, as carrier for LIF were performed by Rittchen et al. [88]. Anti-NG2 antibodies, targeting oligodendrocyte precursor cells (OPCs), were attached on the avidin coated NPs. LIF-loaded PLGA NPs were characterized by scanning electron microscopy on a size (126 ± 50 nm) and morphological levels before performing biological assays. In vitro studies attested of the targeting efficiency toward OPCs, and of LIF activity. To evaluate the CNS remyelination properties of NG2-targeted LIF-NPs, in vivo assays were performed on a mouse brain model having received a stereotaxic injection of myelin toxin, lysophosphatidylcholine, into the Corpus callosum. Remyelination was quantified using electron microscopy by measuring the percentage of myelinated axons and myelin thickness. Overall, NG2-targeted LIF-PLGA NPs induced maturation of OPCs to myelin-competent oligodendrocytes within three days and promoted high-quality myelin repair.

#### 3.5.3. Proteolipid Protein (PLP139–151)

In 2020, Ferreira Lima et al., designed antigen (PLP139–151) loaded PLGA NPs using the double emulsion solvent evaporation method to induce antigen-specific tolerance, regulate immune responses, as well as withstand multiple immunogenic challenges in a MS model. Indeed, this protein was identified as a potent tolerogenic molecule in MS disease. Immunization with this peptide induced experimental autoimmune encephalomyelitis (EAE) in an animal model of MS. Various PLP loaded NPs were prepared with an average diameter of 195.1 ± 10.1 nm for a drug loading of 20%. However, a low encapsulation efficiency (over 60% of PLP not encapsulated) was observed, as well as a complete drug release within the first hours requiring formulation improvements. NPs were incorporated into polymeric microneedles to design a minimal invasive administration system. However, no biological evaluation of the neuroprotective properties of PLP encapsulated PLGA NPs were performed, requiring further studies to confirm their interest.


pharmaceutics-13-01042-t002_Table 2Table 2Non-exhaustive summary of neuroprotective drugs vectorized by the PLGA NP approach for NDD treatment (D.L.: Drug Loading, E.E.: Encapsulation Efficiency).Active SubstanceEncapsulation MethodNP PropertiesModel of UseNeuroprotective ActivitiesRef.Alzheimer’s DiseaseRosmarinic acidEmulsion/solvent diffusionø: 60 to 80 nmPdI: 0.047ζ potential: 2 to 8 mVE.E.: 25 to 40%In vitro: HBMECs, Has and Aβ-insulted SK-N-MC cellsEliminate peroxynitrite anionsReduce inflammatory responsesDisrupt the integrity of HBMEC cytoskeletal microfibrilsDecrease the structural density of TJIncrease the permeability and reduce the TEER by targeting HBMECs with the use of 83-14 Mab[89]Curcumin*Curcuma longa*Single emulsion/solvent evaporationø: 150 to 200 nmζ Potential: −30 to −20 mVIn vitro: GI-1 glioma cellsInhibit Aβ fibril formationReduce brain amyloid level and plaque burdenFree-radical scavenging activity[90]Rhynchophylline *Uncaria rhynchophylla*Nanoprecipitationø: 145.2 nmPdI: 0.133D.L.: 10.3%E.E.: 60%In vitro: bEnd.3 and PC12 cellsIn vivo: healthy C57BL/6 miceInhibit soluble Aβ-induced hyperexcitability of hippocampal neuronsReduce cell apoptosis significantly[58]QuecertinDouble emulsion/solvent evaporationø < 145 nmIn vitro: human neuronal cells (SH-SY5Y)In vivo: BALB/c miceInhibit AChE and secretase enzymesDisrupt amyloid aggregatesPrevent and inhibit Aβ_1_-_42_ fibrils formationDissolve Aβ_1_-_42_ aggregates[91]iA β5Nanoprecipitationø: 153 to 166 nmPdI: 0.090 to 0.100ζ potential: −10.1 to −13 mVE.E.: 63 ± 9%Porcine brain capillary endothelial cellsN/A[92]PhytolEmulsion/solvent evaporationø: 177.4 ± 5.9 nmζ potential: −32.8 ± 2.2 mVD.L.: 56%E.E.: 92%In vitro: Neuro2a cellsIn vivo: healthy male Wistar rats, rats injected with scopolamineAnti-cholinesterase and anti-oxidativeDisrupt amyloid aggregatesAttenuate the impairment of learning and memory induced by scopolamine in rat brainPrevent oxidation of proteins and lipids against scopolamine-induced oxidative damage[93,94]ThymoquinoneSingle emulsion/solvent evaporationø: 226.3 ± 4.6 nmPdI: 0.143ζ potential: −45.7 ± 2.6 mVE.E.: 69.5 ± 3.0%In vivo: STZ-induced Alzheimer model miceAct on neurosis, ROS formation, free radical scavenging capacity, etc.Prevent Aβ aggregates and hyperphosphorylated τ-protein tangles[53]*Nigella sativa* oil and Plasmid DNAModified solvent diffusionø: 600 to 700 nmPdI: 0.229ζ potential: + 35.6 mVD.L.: 13.0%E.E.: 73.8%In vitro: murine neuroblastoma (N2a) cellsInhibit radicals O_2_Promote neurite outgrowth and neuroregeneration[95]Anthocyanins Natural pigmentsEmulsion/solvent evaporationø: 165 nmPdI: 0.400ζ potential: −12 mVE.E.: 60%In vitro: human neuronal cells (SH-SY5Y)Inhibit Aβ_1–42_-induced ROS generationAnti-oxidant[96]Huperzine AEmulsion/solvent evaporationø: 78.1 to 153.2 nmPdI: 0.182 to 0.229ζ potential: −21.2 to +35.6 mVD.L.: 5 to 25%E.E.: 83.2 to 73.8%In vitro: 16HBE and SH-SY5Y cellsIn vivo: KM miceInhibit acetylcholinesteraseDisrupt amyloid aggregates[55]Selegiline Methamphetamine derivativeEmulsion/solvent evaporationø: 217 to 246 nmPdI: 0.295 to 0.331ζ potential: 34.7 to 38.0 mVE.E.: 3.3 to 11.7%In vitro: fibrils (fAβ_1-40_ and fAβ_1-42_)Inhibit MAO-B and Aβ fibrils formationIncrease the destabilizing effects of the drug by increasing selegiline concentration and incubation time[97]
**Parkinson’s Disease**
Schisantherin AFlash nanoprecipitationø: 70.6 ± 2.2 nmPdI: 0.104 ± 0.012ζ potential: −24.7 ± 3.5 mVD.L.: 28.0 ± 0.8%E.E.: 91.1 ± 2.6%In vitro: SH-SY5Y and MDCK cellsIn vivo: larval zebrafish, MPTP-induced zebrafish and male Sprague-Dawley ratsReduce tyrosine hydroxylaseTH+ DA neuronal deathPrevent MPTP-induced decrease in TH+ regionReverse locomotor deficiencyStronger neuroprotective effects in MPP+ -induced SH-SY5Y cell injury model[98]PuearinAntisolvent precipitationø: 88.4 ± 1.7 nmPdI: 0.047 ± 0.007ζ potential: −18.9 ± 2.8 mVD.L.: 43.0 ± 1.6%E.E.: 89.5 ± 1.7%In vitro: MDCK and SH-SY5Y cellsIn vivo: zebrafish and their embryos, male Sprague−Dawley (SD) rats, maleC57BL/6 mice (with MPTP group)Reduce tyrosine hydroxylaseTH+ DA neuronal deathReduce behavioral deficits and associated mobility impairments in MPTP-mediated neurotoxicity in miceReduce TH+ neuron loss and associated neurotoxicity[63]LevodopaDouble emulsion/solvent evaporationø: 329.0 to 383.7 nmPdI: 0.384 to 0.426ζ potential: −4.5 to −20.8 mVE.E.: 50.5 to 73.0%In vitro: PC12 cellsIn vivo: healthy CD57/BL6 mice (with MPTP group)Improve locomotor activity over timeImprove brain delivery by intranasal routeEnhance dopamine concentration in the brain compared to the blood[99]RasagilineDouble emulsion/solvent evaporationø: 221.7 ± 5.7 nmPdI: 0.388 ± 0.860ζ potential: −36.1 ± 4.4 mVE.E.: 29.2 ± 1.8%In vivo: healthy and Parkinson Wistar ratsInhibit MAO-B enzymePrevent neuronal damage caused by oxidative stress[67]
**Huntington’s Disease**
Peptides QBP1, NT17, and PGQ_9_P^2^Nanoprecipitationø: 158 to 180 nmPdI: 0.031 to 0.066ζ potential: −23.3 to −27.5 mVIn vitro: MDCK, Neuro 2A and PC12 cell model of HDIn vivo: healthy mice, Drosophila, larvae, and adult fly model of HDInhibit mHtt aggregationRestore motor activity[74]CholesterolNanoprecipitationø: 200 to 300 nmPdI: 0.090 to 0.300ζ potential: −8 to −12 mVD.L.: 0.7 to 2.5%E.E.: 48 to 68%In vitro: neural stem (NS) cell model of HDIn vivo: genotyping of R6/2 mouseRestore synaptic alterations and delaying cognitive[77]
**Amyotrophic Lateral Sclerosis**
PHA-767491Nanoprecipitationø: 141 to 155 nmPdI ≤ 0.150D.L.: 24%E.E.: 12 to 18%In vitro: SH-SY5Y cell model of ALSReduce TDP-43 phosphorylation[81]
**Multiple Sclerosis**
Interferon-β-1aDouble emulsion/solvent evaporationø: 150 to 165 nmPdI: 0.081 to 0.093ζ potential: 17.7 to 18.8 mVD.L.: 0.7 to 2.5%E.E.: 95.9%In vitro: human blood plasma, primary hepatocytes from male Wistar ratsIn vivo: Wistar male ratsN/A[85]LIFDouble emulsion/solvent evaporationø: 126 ± 50 nmIn vitro: oligodendrocyte precursor cells from neonatal Sprague–Dawley ratsInduce maturation of OPCs to myelin-competent oligodendrocytes within 3 daysPromote high-quality myelin repair[88]Proteolipid ProteinDouble emulsion/solvent evaporationø: 195.1 ± 10.1 nmPdI: 0.220ζ potential: −20 mVD.L.: 20%E.E.: 40%N/AN/A[100]


## 4. New Candidates for PLGA NPs Vectorization

Many active substances with neuroprotective activities remain to be evaluated for the PLGA NP approach to improve their brain delivery [101,102,103,104,105,106,107,108,109,110,111,112,113,114,115]. Some of them, such as trehalose, exhort numerous properties ranging from anti-inflammatory to anti-oxidant, which can potentially be applied for out of current applications and NDDs treatment. Trehalose is a non-reducing disaccharide that has been widely used in the food industry (Figure 7). Still, it has also recently shown many unique properties, indicating its potential use in preventing neurodegeneration [105]. It may act as a potent stabilizer of proteins, preserving protein structural integrity and, as an mTOR-independent autophagy inducer, improving the clearance of the mutant proteins that act as autophagy substrates when aberrant protein deposition occurs. Trehalose treatment reduced the level of toxic protein aggregates, which in turn improve behavioral symptoms and survival in animal models of NDDs including AD [106], PD [107,108], and HD [109]. However, trehalose can be cleaved by trehalase, an enzyme observed in different mammalian organs [110]. Therefore, PLGA NPs could be interesting candidates to protect trehalose from degradation and improve its bioavailability.

Another good example is the association of salvianolic acid B (Sal B), tanshinone IIA (TS IIA), and panax notoginsenoside (PNS) (Figure 7), which demonstrated neuroprotective effects on cerebral ischemia reperfusion (I/R) injury. In 2013, Zhang and coworkers encapsulated them into PLGA NPs for brain delivery via the intratympanic route (inner ear administration) [111]. Homogeneous suspensions were obtained by double emulsion solvent evaporation method and composed of particles with a mean diameter of 154 nm. In vitro, loaded PLGA NPs exhibited an excellent sustained release of 76% and 93% of the drug after 72 h for Sal B and PNS, respectively. A slower release was observed for TS IIA (23.6% after 72 h), explained by the drug low solubility in the medium. Pharmacokinetic studies revealed that PLGA NPs improved drug distribution within the inner ear, the CSF, and the brain after intratympanic administration demonstrating the usefulness of this route to deliver drugs without BBB crossing. The anti-oxidant activity of such nanosystems was then evaluated since recent studies confirmed that oxidative stress (SOD, MDA, NOS, ROS and NO) plays a role in the pathogenesis of I/R injury. The lipid peroxidation evaluation showed that drug-loaded PLGA NPs exhibited neuroprotection by regulation of SOD, MDA, and NOS levels, especially a decrease of MDA and NOS levels validating the anti-oxidant properties of Sal B, TS IIA, and PNS. Combining these compounds with PLGA NPs may be a very interesting new tool for brain disease treatment, such as AD and PD thanks to their neuroprotective activity. Indeed, Young Woo Lee and coworkers demonstrated that Sal B improved memory impairment caused by Aβ_25–35_ peptide-induced neuronal damage in AD [103] while Zeng et al., published its ability to protect cells against MPP-induced apoptosis for PD treatment [112]. TS IIA inhibited transcription and translation of iNOS, MMP-2, and NF-κBp65, and suppressed expression of NADPH oxidase and iNOS, respectively, involved in AD [113] and PD [114] development. Finally, in AD, PNS Rb1 suppressed the phosphorylated tau protein expression and upregulated the expression levels of brain-derived neurotrophic factor (BDNF) [115].

## 5. Conclusions

BBB crossing or the potential degradation of neuroprotective substances remain strategic points needing to be overcome for NDD treatments. In this context, PLGA NPs are a promising approach for new therapeutics. Being both biocompatible and biodegradable, they allow for drug protection, enhance their bioavailability, and possess active targeting capabilities facilitating the delivery to a precise site. These beneficial effects were used for active substance administration (i.e., thymoquinone, curcumin, etc.) and delivery directly into the brain without losing their neuroprotective properties. Moreover, they also demonstrated therapeutic activities such as restoration of the lysosomal pH in the case of PD. Some challenges remain regarding the biological (i.e., biological barrier breaching), technological (i.e., scale-up synthesis), and study design (i.e., data treatment) levels, which impact the potential clinical trials of PLGA NPs. They need to be addressed for the successful development of such therapeutics [116]. However, new guidelines are established to facilitate the transposition from bench to clinic and ensure the product quality and safety [21,117].

Even if these researches are often in the early stages of development, nanovectorization of neuroprotective substances announce future great advances in the pharmaceutical field for NDD treatment.

## Figures and Tables

**Figure 1 pharmaceutics-13-01042-f001:**
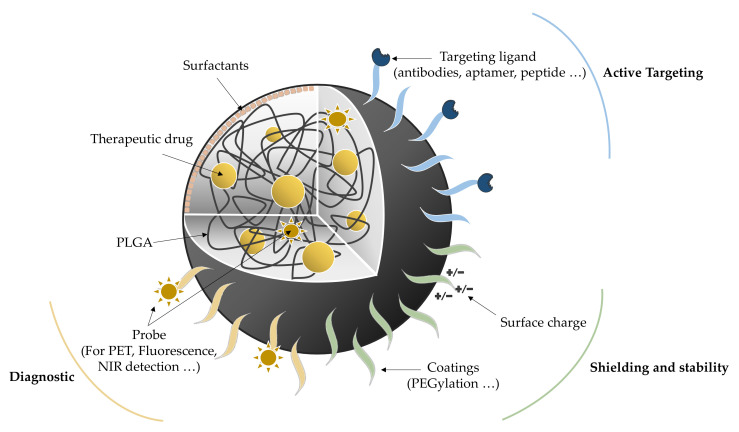
Schematic representation of PLGA NPs and their multiple potential functionalizations: for diagnostic, targeting, shielding, or stability purposes [30] (NIR: Near-infrared; PET: Positron-emission tomography).

**Figure 2 pharmaceutics-13-01042-f002:**
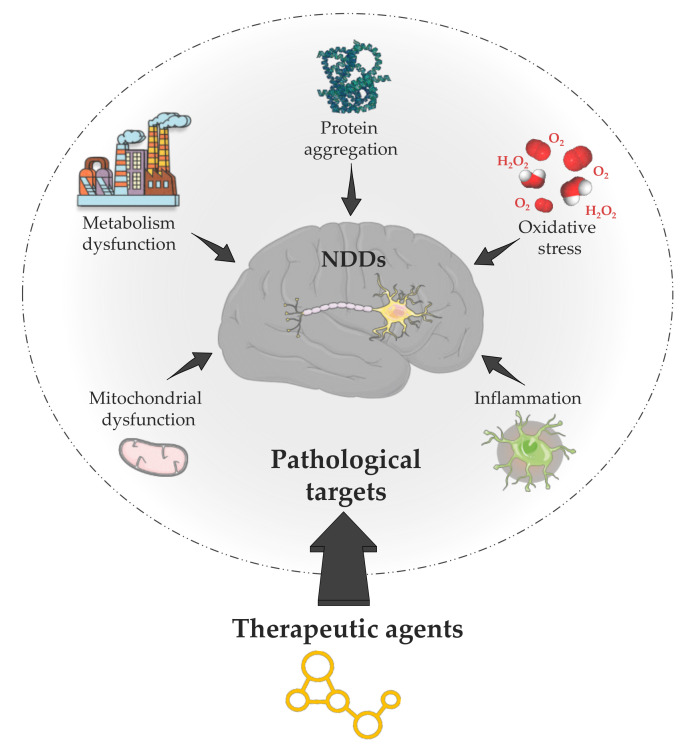
Pathological targets for active substances in the case of the most common NDDs: protein aggregation, oxidative stress, inflammation, mitochondrial, and metabolism dysfunction.

**Figure 3 pharmaceutics-13-01042-f003:**
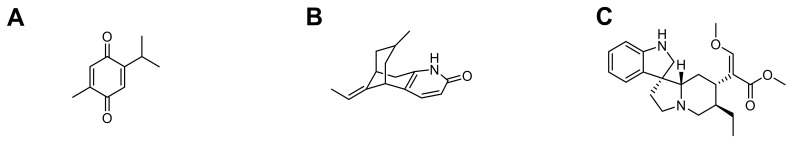
Chemical structures of the three described compounds encapsulated into PLGA NPs for AD treatment (Thymoquinone (**A**), Huperzine A (**B**), and Rhynchophylline (**C**)).

**Figure 4 pharmaceutics-13-01042-f004:**
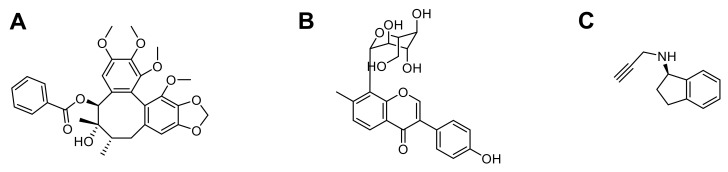
Chemical structures of the three described compounds encapsulated into PLGA NPs for PD treatment (Schisantherin A (**A**), Puerarin (**B**), and Rasagiline (**C**)).

**Figure 5 pharmaceutics-13-01042-f005:**
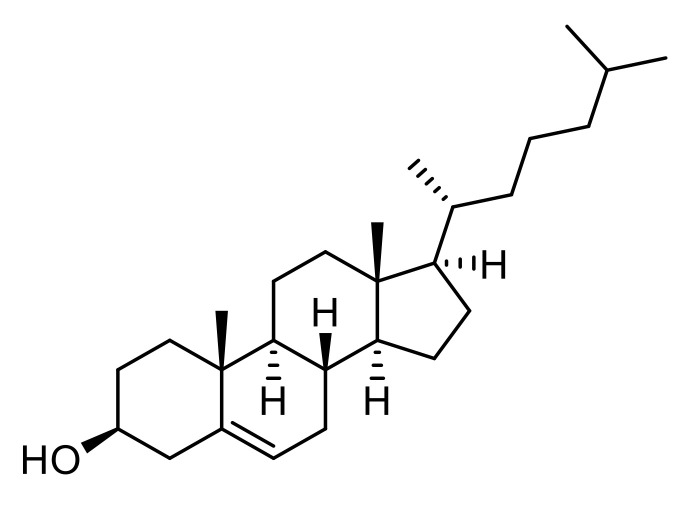
Chemical structure of cholesterol encapsulated into PLGA NPs for HD treatment.

**Figure 6 pharmaceutics-13-01042-f006:**
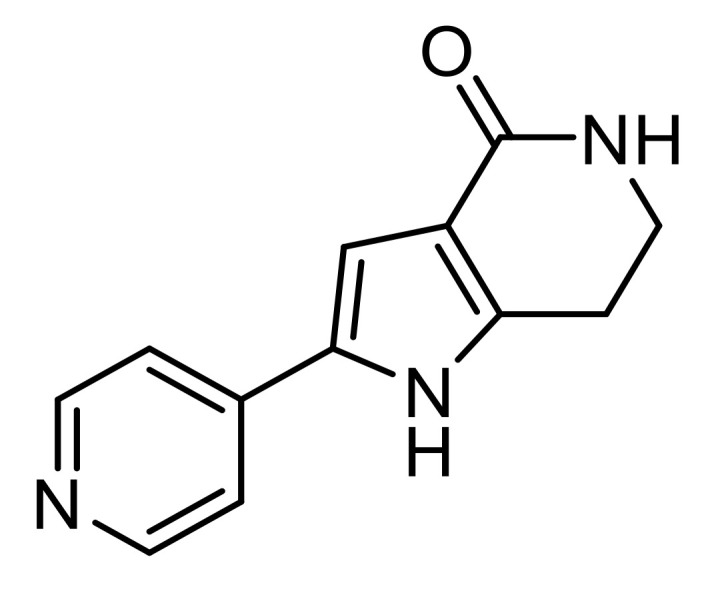
Chemical structure of PHA-767491 encapsulated into PLGA NPs for ALS treatment.

**Figure 7 pharmaceutics-13-01042-f007:**
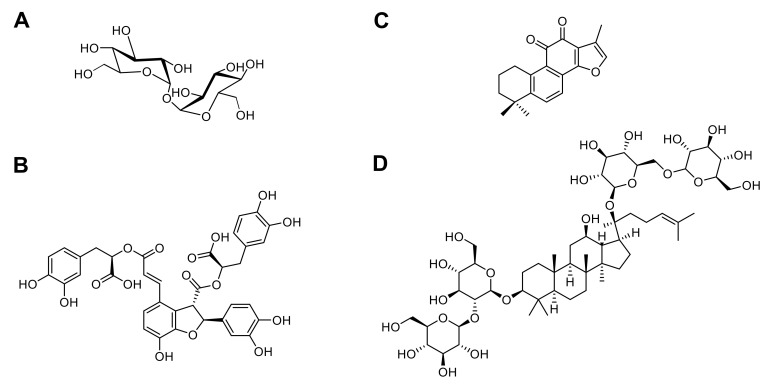
Chemical structures of the four described compounds for AD and PD treatment (Trehalose (**A**), Salvianolic acid B (**B**), Tanshinone IIA (**C**), and Panax notoginsenoside Rb1 (**D**)).

## Data Availability

Not applicable.

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
