# Peer review of "PLGA-Based Nanoparticles for Neuroprotective Drug Delivery in Neurodegenerative Diseases"

_pharmaceutics, 2021, doi:10.3390/pharmaceutics13071042_

Round 1

Reviewer 1 Report

The manuscript by Cunha et al. provided clear and up-to-date summary of PLGA-based NP in neurodegenerative diseases, which might be attractive to readers from multiple fields. I recommend to accept the paper after minor revision.

  1. Page 3, line 92, the authors covered the major properties of NPs, which is good. To add on, the degradation properties of PLGA which affects the controlled release behavior (e.g., molecular weight, ratio of PLA to PGA) should be discussed in the section.
  2. Page 3, line 101: the author addressed the fluorescent probes can be introduced to NPs for positron-emission tomography, or single-photon emission computed tomography. However, for PET and SPECT, the imaging probe is isotope but not fluorescent agent.
  3. Page 3, line 108, the author started mentioning intranasal administration, but there was not background introduction to it. Although intranasal administration is essential to neurodegenerative diseases, quick introduction is needed so readers from different fields would be confused. Or the author can consider to discuss this section after the introduction from line 123 is finished.
  4. In page 4, line 134, the authors stated that there are two strategies for PLGA NPs to treat NDD. However, the manuscript focused on the vectorization, while the content on the inherent therapeutic effect of PLGA NPs was limited. Would the authors provide more information or evidence on it as the topic will be attractive to many readers.
  5. As polysorbate 80 is a key material in PLGA NPs to achieve CNS delivery, I would suggest the authors to briefly extend the discussion on line 106 such as coating technique (or how do they interact to stay stable?), the density of coating, and the potential risk (e.g., safety) upon brain tissue exposure.
  6. PLGA has already been approved by FDA as an excipient. However, there is still little progress of PLGA-based vector in clinical development. It would be great if the authors can address the current obstacles in the pharmaceutical field of PLGA NPs and provide information on the development stage.

Author Response

Reviewer #1

  1. Page 3, line 92, the authors covered the major properties of NPs, which is good. To add on, the degradation properties of PLGA which affects the controlled release behavior (e.g., molecular weight, ratio of PLA to PGA) should be discussed in the section.

Answer: We want to thank the reviewer 1 for this comment. We modified the “PLGA NPs” part to improve its relevance by developing how PLGA degradation is linked to the NP design and composition.

  1. Page 3, line 101: the author addressed the fluorescent probes can be introduced to NPs for positron-emission tomography, or single-photon emission computed tomography. However, for PET and SPECT, the imaging probe is isotope but not fluorescent agent.

Answer: The reviewer 1 is totally right, the imaging probe for PET and SPECT was missing, the manuscript has been modified accordingly.

  1. Page 3, line 108, the author started mentioning intranasal administration, but there was not background introduction to it. Although intranasal administration is essential to neurodegenerative diseases, quick introduction is needed so readers from different fields would be confused. Or the author can consider to discuss this section after the introduction from line 123 is finished.

Answer: As mentioned by reviewer 1, intranasal route is a very interesting option especially in the case of NDD treatment. To facilitate the reader understanding and present this administration route, complementary information has been added in the paragraph 2.2.

  1. In page 4, line 134, the authors stated that there are two strategies for PLGA NPs to treat NDD. However, the manuscript focused on the vectorization, while the content on the inherent therapeutic effect of PLGA NPs was limited. Would the authors provide more information or evidence on it as the topic will be attractive to many readers?

Answer: To comply with the comments, more details about PLGA NP therapeutic effects, in particular its ability to restore lysosomal deleterious effects, were added.

  1. As polysorbate 80 is a key material in PLGA NPs to achieve CNS delivery, I would suggest the authors to briefly extend the discussion on line 106 such as coating technique (or how do they interact to stay stable?), the density of coating, and the potential risk (e.g., safety) upon brain tissue exposure.

Answer: We thank the reviewer 1 for this comment. Indeed polysorbate 80 is a key component in NPs coating to enhance drug delivery directly into the brain. Therefore, we added information to develop the discussion for the reader.

  1. PLGA has already been approved by FDA as an excipient. However, there is still little progress of PLGA-based vector in clinical development. It would be great if the authors can address the current obstacles in the pharmaceutical field of PLGA NPs and provide information on the development stage.

Answer: These aspects are very interesting points. To date, no clinical trials using PLGA NP are reported, only pre-clinical studies. The corresponding information was added to the manuscript to improve it, including some aspects linked to the current limitations of industrial scale up in the conclusion.

Reviewer 2 Report

The review of Cunha et al. entitled "PLGA-based nanoparticles for neuroprotective drug delivery in neurodegenerative diseases” presents an overview of the lasted findings, and advances in PLGA nanoparticles for neuroprotective drug delivery in the case of neurodegenerative disease treatment.  It is a well written manuscript and the topic is very pertinent and novel. Although some interesting data have been presented, several issues need to be addressed and added before this manuscript could be considered for publication. I would recommend a resubmission with major revision based on the following general comments:

  1. It is missed a chapter related to the different methodologies to prepare PLGA nanoparticles and the required reactants. During PLGA synthesis, it is used many organic solvents and surfactants that can generate drawbacks during in vitro/in vivo experiments. It is important to mention the pros and cons of each synthetic route and its potential scalability. PLGA nanoparticles have a high surface area and their production and clean is problematic at large quantities. On the other hand, it should be mentioned that PLGA present low encapsulation yields for many drugs, and the ones encapsulated in the PLGA outside shell can be delivered not at the target site.
  2. In general terms, from a commercial point of view, it is important to mention if there is any PLGA nanoparticles approved by FDA or EMA for any neurodegenerative disease treatment, or the number of patents related to PLGA nanoparticles and neurodegenerative disease.

Line 94. According to literature, a moderate stability in nanoparticles is achieved at zeta potential >+30 mV or <-30mV (https://doi.org/10.1016/B978-0-08-100557-6.00003-1).

Lines 99-135. These paragraphs are a bit confused and messy. It can be split into parts, first, about PLGA NP applications and second, about the administration routes.

Line 376. Cholesterol molecule is wrong.

Author Response

Reviewer #2

  1. It is missed a chapter related to the different methodologies to prepare PLGA nanoparticles and the required reactants. During PLGA synthesis, it is used many organic solvents and surfactants that can generate drawbacks during in vitro/in vivo experiments. It is important to mention the pros and cons of each synthetic route and its potential scalability. PLGA nanoparticles have a high surface area and their production and clean is problematic at large quantities. On the other hand, it should be mentioned that PLGA present low encapsulation yields for many drugs, and the ones encapsulated in the PLGA outside shell can be delivered not at the target site.

Answer: We can understand the comment of reviewer 2. PLGA synthesis and scalability are two crucial aspects for NP formulation and the potential use for biomedical applications. However, in this review, we have chosen not to develop this point thoroughly, which is already well-reviewed, but to focus on a different approach (less reviewed), namely the encapsulation of API for NDD treatment. We added information in the manuscript and referred to some very interesting papers to help the reader on the PLGA nanoparticles formulation aspect.

Regarding the low encapsulation yield and the potential off-target delivery, we made modifications to improve the manuscript.

  1. In general terms, from a commercial point of view, it is important to mention if there is any PLGA nanoparticles approved by FDA or EMA for any neurodegenerative disease treatment, or the number of patents related to PLGA nanoparticles and neurodegenerative disease.

Answer: The comment of reviewer 2 is very interesting about this approach's commercial point of view. However, to date, no PLGA nanoparticles are approved by the regulatory agencies. They are only at the preclinical stage; some factors need to be optimized for a potential industrial scale-up. To address this point, we added this information to the manuscript.

  1. Line 94. According to literature, a moderate stability in nanoparticles is achieved at zeta potential >+30 mV or <-30mV (https://doi.org/10.1016/B978-0-08-100557-6.00003-1).

Answer: We want to thank reviewer 2 for this comment. To comply with the current guidelines, we modified the zeta potential values as mentioned by reviewer 2.

  1. Lines 99-135. These paragraphs are a bit confused and messy. It can be split into parts, first, about PLGA NP applications and second, about the administration routes

Answer: We are sorry if reviewer 2 finds this specific part quite confusing and messy. We tried our best to improve the understanding of this part, notably by splitting it as proposed by reviewer 2.

  1. Line 376. Cholesterol molecule is wrong.

Answer: We thank the referee for this observation. Indeed, figures 5 and 6 were exchanged in the manuscript. The modification was made accordingly.
